# What's governance got to do with it? Examining the relationship between governance and deforestation in the Brazilian Amazon

Rayna Benzeev[1]*, Bradley Wilson[2], Megan Butler[3], Paulo Massoca[4], Karuna Paudel[5], Lauren Redmore[6¤], Lucía Zarbá[7]

1 Department of Environmental Studies, University of Colorado Boulder, Sustainability, Energy, and Environment Complex, Boulder, Colorado, United States of America, 2 First Street Foundation, Brooklyn, New York, United States of America, 3 Lake Superior State University, Sault Ste Marie, Michigan, United States of America, 4 Paul O'Neill School of Public and Environmental Affairs (SPEA) and Center for the Analysis of Social-Ecological Landscapes (CASEL), Indiana University, Bloomington, Indiana, United States of America, 5 Warnell School of Forestry and Natural Resources, University of Georgia, Athens, Georgia, United States of America, 6 Recreation, Park and Tourism Sciences Department, Texas A&M University, College Station, Texas, United States of America, 7 Instituto de Ecología Regional (IER), Universidad Nacional de Tucumán (UNT) - Consejo Nacional de Investigaciones Científicas y Técnicas (CONICET), Tucumán, Argentina

¤ Current address: Aldo Leopold Wilderness Research Institute, Rocky Mountain Research Station, USDA Forest Service, Missoula, Montana, United States of America
* rayna.benzeev@colorado.edu

**Data Availability Statement:** Our data set was assembled from freely and openly available sources including the the national repository of

## Abstract

Deforestation continues at rapid rates despite global conservation efforts. Evidence suggests that governance may play a critical role in influencing deforestation, and while a number of studies have demonstrated a clear relationship between national-level governance and deforestation, much remains to be known about the relative importance of subnational governance to deforestation outcomes. With a focus on the Brazilian Amazon, this study aims to understand the relationship between governance and deforestation at the municipal level. Drawing on the World Bank Worldwide Governance Indicators (WGI) as a guiding conceptual framework, and incorporating the additional dimension of environmental governance, we identified a wide array of publicly available data sources related to governance indicators that we used to select relevant governance variables. We compiled a dataset of 22 municipal-level governance variables covering the 2005–2018 period for 457 municipalities in the Brazilian Amazon. Using an econometric approach, we tested the relationship between governance variables and deforestation rates in a fixed-effects panel regression analysis. We found that municipalities with increasing numbers of agricultural companies tended to have higher rates of deforestation, municipalities with an environmental fund tended to have lower rates of deforestation, and municipalities that had previously elected a female mayor tended to have lower rates of deforestation. These results add to the wider conversation on the role of local-level governance, revealing that certain governance variables may contribute to halting deforestation in the Brazilian Amazon.

electoral results (TSE), the Brazilian Institute of Geography and Statistics (IBGE), the Brazilian Amazon satellite deforestation monitoring program (PRODES), the Chico Mendes Institute of Biodiversity Conservation (ICMBio), and the Ministry of Environment (MMA). All data sources can be re-acquired from their original sources without any special access privileges. Links to the individual sources for each variable are listed in the glossary section of the SI. All data pre-processing steps required to reproduce our results are described in detail in section 2.3 in the manuscript. Code to reproduce the model results is hosted at: https://github.com/bradleyswilson/governance_deforestation (DOI: 10.5281/zenodo.6600179).

**Funding:** This work was supported by the National Science Foundation DBI-1052875 to the National Socio-Environmental Synthesis Center. PM was funded by the Brazilian Science without Borders program (ScF/CNPq 234533/2014-5).

**Competing interests:** The authors have declared that no competing interests exist.

## 1. Introduction

Reducing deforestation is one of the most promising and cost-effective solutions to mitigate climate change and respond to the biodiversity crisis [1–4]. However, the world's forests continue to diminish at high rates, particularly across the tropics [5, 6], driven by biophysical, socioeconomic, institutional, and political factors teleconnected across diverse geopolitical scales [7–9]. Increasingly, scholars and development organizations alike point to governance, or the interactions of diverse agents in devising institutions that shape behavior and influence both decision-making processes and outcomes [10], as a critical factor influencing forest outcomes [e.g. 11–13]. Forest governance–defined as "the set of regulatory processes, mechanisms and organizations through which political actors influence forest actions and outcomes" [13]– occurs across multiple spatio-temporal levels and scales, involving interactions between actors with different incentives, responsibilities, and practices related to use, management, and protection of forest areas and resources. Governance is not synonymous with government, though government does play a role in governance [14]. Governance has been recognized as an underlying cause of deforestation by indirectly influencing the direct (proximate) drivers of deforestation (e.g. agricultural expansion) [7, 15, 16]. However, there are no analytical outcome-oriented standards for defining what "good" governance entails, particularly for specific aspects of good governance, in relation to deforestation [16, 17].

Globally, many governments have devolved at least partial responsibility for forest management, monitoring, and protection to subnational levels [18]. Research suggests that subnational levels of government may have enough governance authority to influence forest conservation [19]. Decentralization has allowed for a shared approach from local to international levels of governance to address the context-specific realities of complex and dynamic socio-environmental forest systems [20–23]. In countries where forest legislation is primarily produced at the federal level, subnational levels, including states and municipalities, have often been responsible for mediating how laws and policies are interpreted and enforced on the ground [24, 25]. As a result, forest governance may vary greatly across local levels [26].

Relatively little research has focused on the impact of municipal-level governance on forest change, despite evidence that local-level governance is important and should be monitored by policymakers [15, 16, 26, 27]. Most comparative quantitative studies that analyzed the impact of governance on forest cover focused on national-level governance [e.g. 28–30]. Studies at the municipal level have primarily been case studies examining governance processes that are difficult to standardize and compare across a large sample of municipalities [31, 32]. Only one study we are aware of conducted a cross-municipal analysis of deforestation outcomes and governance in Brazil, though no clear relationships were found [33]. The abundance of research on governance and deforestation from a cross-national perspective, which has provided context for the aspects of governance that matter most, highlights the notable gap in governance research from a cross-*municipal* perspective.

There is also a need to better understand the relationships between different components of governance and deforestation [16]. Although several studies found that stronger governance often related to reduced deforestation [30, 34], individual governance indicators have had different and sometimes opposite relationships to deforestation and other environmental factors. Several governance indicators have been linked to positive outcomes for forests and the environment. For example, voice and accountability, the ability of citizens to democratically influence policy, has been associated with positive environmental outcomes [29, 35, 36]. Factors such as participation and the strength of democratic institutions, which represent accountability and transparency in both informal and formal rules, have positively influenced countries' abilities to achieve sustainable development goals [37]. Strongly democratic countries have

been shown to have less deforestation than weakly democratic countries, as weaker democracies have often allowed forests to be exploited [38]. The quality of public services provided by local governments, often considered an indicator of good governance, has been linked to environmental protection [39, 40]. Both environmental governance and governments' abilities to create fair and predictable rules through rule of law have also been shown to relate to more sustainable forest outcomes [34, 41, 42].

Other studies have found that some indicators of governance were correlated with negative outcomes for forests and the environment. For example, in some situations where good governance reduced bureaucratic challenges facing private businesses, good governance was also associated with negative environmental outcomes, including higher deforestation [43, 44]. Furthermore, stronger democracy and political rights, including electoral process, political pluralism, and the protection of individual rights, have been associated with higher deforestation rates in areas with popular support for industrialization, resource extraction, and land use change [34].

For some governance indicators, the expected relationship with deforestation is still unclear due to mixed findings across multiple studies. One study found that strong regulatory quality, or governments' abilities to create sound policies for ease of private sector growth, was correlated with negative environmental outcomes [45], whereas another study found that weak regulatory quality was correlated with negative environmental outcomes [46]. Political stability, which ensures continuity of policies over time, was found to result in both positive [47, 48] and mixed environmental outcomes [49]. While one study found that corruption was strongly associated with the expansion of agricultural and cattle operations, resulting in increases in deforestation [50], another study found that countries with more corruption had more forest cover [48]. These mixed findings indicate that specific governance indicators may have varying relationships with forest and environmental outcomes depending on the local context [34, 51, 52].

In this study, we pulled from over a decade of publicly available and standardized data for municipalities across the Brazil Amazon to ask: What is the relationship between governance and deforestation at the municipal level? Our interdisciplinary team explored this relationship for 22 variables representing five governance indicators across 457 municipalities from 2005 to 2018. This study contributes to the wider conversation on the extent to which subnational governance relates to deforestation outcomes. Considering recent calls to synthesize publicly available data as part of novel research studies, we also aimed for our interdisciplinary methods to serve as a roadmap to integrate local-level social and environmental data to answer questions of global conservation importance.

## 2. Research design

We developed our research using a collaborative, interdisciplinary approach throughout research design, analysis, and interpretation. We iteratively assembled a theoretically grounded dataset of governance-relevant variables for use in a panel analysis of municipal-level governance and deforestation in the Brazilian Amazon. Below we describe our study region, framework development, data preparation, and model specification.

### 2.1 Study region and context

Our study included all municipalities in the Amazon biome in Brazil for which data on deforestation were available from 2005–2018 through the official monitoring system of Brazil (PRODES) (n = 457, Fig 1). The Amazon biome in Brazil intersects with nine states, spanning an area of 4.2 million km$^2$ [53]. The biome contains some of the highest known levels of

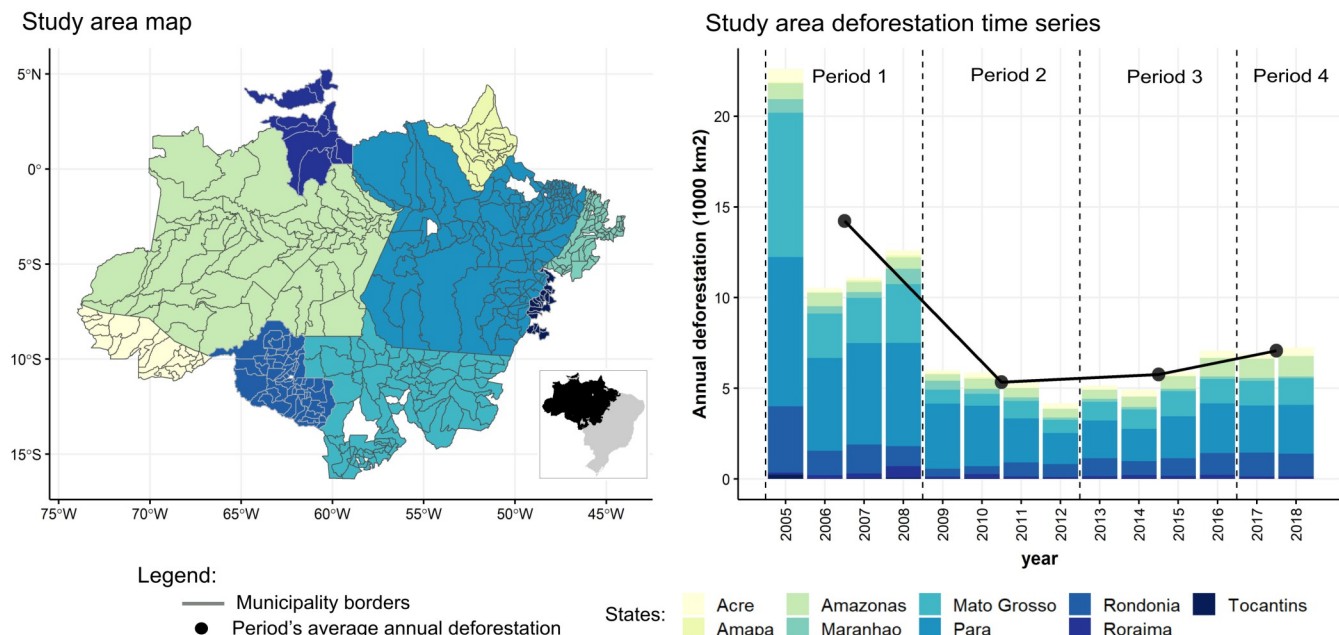

**Fig 1. Study area map and deforestation time series.** Left: The 457 municipalities analyzed in the study. Right: A time series of the total annual deforestation in the study area. Colors represent states. Black circles represent average yearly deforestation for the four time periods, which were used in calculating the dependent variable in the analysis.

biological diversity on earth and diverse groups of people inhabit the area, including indigenous and forest-dependent populations, making the Amazon a rich mosaic of biological, ecological, and socio-cultural diversity [54]. Approximately 80% of the primary forest area remains standing today, with much of the remainder converted to agriculture [53]. The conversion of the Amazon to agriculture is widely perceived to be a threat to global climate change and sustainable development targets, alike, and governance is seen by many as a key factor to accelerate or decrease forest loss [55–59].

In 1988, Brazil's Federal Constitution implemented a tiered public management system whereby forest governance responsibilities were shared across municipalities, states, and the federal government [60, 61]. However, the federal and state governments have remained the major players in designing, implementing, and enforcing forest regulations [62, 63]. Municipal governments have abided by national laws when devising and implementing subnational rules and programs, but in many cases they also strengthened more local forms of forest governance. The diverse array of local-level programs and initiatives repositioned municipalities as key players in tackling deforestation in the Amazon [64, 65]. In 2006, for instance, the municipality of Lucas do Rio Verde (Mato Grosso state) devised an innovative program to monitor local land use and land cover changes (*Lucas do Rio Verde Legal*), including pioneering a system to geocode and register landholdings [66]. Likewise, in 2008 the municipality of Paragominas (Pará state) devised a set of collective arrangements led by the mayor, local farmers' and rural producers' unions, and external NGOs, to halt deforestation rates and to enter the geocoded information of landholdings into a public registry [61]. Supported by both state and federal governments and in cooperation with external funding agencies and NGOs, municipalities in the Brazilian Amazon have received increased assistance in structuring and equipping municipal agencies and in training local agents. The increased number of programs and support

targeting the municipal level have broadened the scope of municipal environmental agendas, including greater participation in enforcing forest regulations [67–70].

The List of Priority Municipalities (LPM) represented a critical policy focused on municipal-level environmental governance in the Brazilian Amazon. Implemented by the Ministry of Environment in 2007 and considered to be a central tenet of the 2004 federal Action Plan for Prevention and Control Deforestation in the Amazon (PPCDAm), the LPM policy targeted municipalities considered deforestation hotspots in the region. Mayors and other local stakeholders were required to cooperate and coordinate actions to comply with targets for both reducing deforestation and registering property boundaries for deforestation monitoring. However, the performance of municipal governments in governing forest resources and tackling activities related to deforestation varied greatly across diverging context-specific realities, with some municipalities taking significant action and others taking very little [70]. Even so, the LPM contributed substantially to the drastic reduction in deforestation rates that occurred in the Amazon from 2004 to 2012 (Fig 1) [62, 71–74].

## 2.2 Methods

**2.2.1 Phase one: Governance framework development and data collection.** The development of analytical governance frameworks has been instrumental for researchers and organizations to understand and systematically compare important characteristics of governance systems across diverse localities [e.g. 34, 51, 75–77]. To develop the framework used in this analysis, we drew on the World Bank's Worldwide Governance Indicators (WGI) framework as a starting point to select governance indicators [78]. Many frameworks have been developed and operationalized to advance understanding of the role of governance in environmental management, including Program on Forests [75], the World Resources Institute [76], and the International Union for the Conservation of Nature [79], among others. We chose the WGI framework to guide our study because it is widely used by practitioners and policymakers in the field of international development [80]. We are therefore able to enter a global conversation with implications for policy at scale. The framework consists of a set of six indicators of governance: Voice and Accountability, Political Stability and Absence of Violence, Government Effectiveness, Regulatory Quality, Rule of Law, and Control of Corruption. This framework was proposed for measuring national-level governance based on perceptions data through surveys to independent organizations and parties. Given that our specific research goals did not include original data collection, but rather a synthesis of publicly available data, we adapted the framework as described below.

The WGI framework uses public perceptions data to measure each indicator; however, these data, to our knowledge, are available at the national level and not at the municipal level. As such, we were unable to use a similar perceptions-based dataset, and we therefore relied on publicly available reported data representing proxies of governance outcomes. We collected longitudinal data from various Brazilian government-sponsored data sources, including the national repository of electoral results (TSE), the Brazilian Institute of Geography and Statistics (IBGE), the Brazilian Amazon satellite deforestation monitoring program (PRODES), the Chico Mendes Institute of Biodiversity Conservation (ICMBio), and the Ministry of Environment (MMA). In total, we identified over 105 potential variables from 17 sources (S1 Table) that tracked changes in governance across a wide array of sectors, including public policy, law, commercial enterprises, and the environment, among others. We then trimmed this initial larger dataset to fit within the constraints of our analysis. We examined the definitions and data collection processes of each variable to identify which ones most closely aligned with each indicator definition. We then assigned relevant variables to each indicator category. We

**Table 1. Indicators, definitions, and hypothesized relationships between each indicator and deforestation for the governance analytical framework.** The term "Positive" indicates an association with increased deforestation, the term "Negative" indicates an association with reduced deforestation, and the term "Unclear" indicates that the relationship is uncertain. All indicator definitions were adapted from Kaufmann (1999) except Environmental Governance, which was sourced from Lemos and Agrawal (2006) [14, 78].

| Governance indicator | Indicator definition | Related studies and relationship with deforestation | Hypothesized relationship with deforestation |
|---|---|---|---|
| **Voice & Accountability (VA)** | The extent to which citizens are able to participate in selecting their government, as well as freedom of expression, freedom of association, and a free media. | *Positive*: Shandra (2007) | Unclear |
| | | *Negative*: Wehkamp et al. (2018), Ehrhardt-Martinez et al. (2002), Shandra et al. (2009) | |
| | | *Unclear*: Mejía Acosta (2013) | |
| **Regulatory quality (RQ)** | The ability of the government to formulate and implement sound policies and regulations that permit and promote private sector development. | *Positive*: Barbier & Tesfaw (2015), Huang et al. (2018) | Positive |
| **Rule of law (ROL)** | The extent to which agents abide by the rules of society, and in particular the quality of contract enforcement, property rights, the police, and the courts, as well as the likelihood of crime and violence. | *Negative*: Wehkamp et al. (2018) | Negative |
| | | *No correlation*: Abman (2018) | |
| **Government effectiveness (GE)** | The quality of public services, the quality of the civil service and the degree of its independence from political pressures, the quality of policy formulation and implementation, and the credibility of the government's commitment to such policies. | *Negative*: Contreras-Hermosilla (2011), Park et al. (2007) | Negative |
| **Environmental governance (EG)** | The local regulatory processes, rules, and mechanisms and organizations used to influence environmental outcomes. | *Negative*: Nepstad et al. (2014), Wehkamp et al. (2018), Shandra et al. (2009) | Negative |

retained only a subset of the initial set of variables, selecting those that were relevant to the governance indicators. We removed those that were poorly representative of governance concepts, those that varied so significantly between years or across municipalities that we had reason to suspect errors, and those with a narrow temporal window (see section 2.3.1). This iterative process guided us to select a modified framework of five governance indicators (Table 1). Ultimately, we had to withdraw two of the WGI indicators, control of corruption and political stability, due to a lack of municipal-level data. In addition, we incorporated the indicator Environmental Governance to specifically assess the role of local regulatory processes, rules, and mechanisms and organizations used to influence environmental outcomes [14]. Our decision to measure Environmental Governance is supported by several studies [33, 41, 43, 51, 81]. Using this modified framework, we conducted a review of previous studies to determine hypothesized relationships between each indicator and deforestation (Table 1).

We recognize that there is a distinction between the concept of governance, indicators of governance, and reported data that serve as proxies for governance indicators [33]. Studies have suggested that governments are more likely to measure demographic statistics or day-to-day activities of governments rather than the progress or outcomes produced by these activities [82]. This may limit the extent to which government-tracked data represents governance processes. Although we relied on official government-sponsored surveys and census data in this analysis, our representation of governance systems and outcomes is imperfect. We consider the implications of data-related challenges throughout the discussion.

## 2.3 Phase two: Data preparation

**2.3.1 Independent variables: Municipal-level governance predictors.** During the timeframe of the study, relatively few municipal-level data sources were available annually, since many surveys did not collect data on the same survey questions and themes in consecutive years. To account for these discrepancies, we aggregated variables in the final dataset into

**Table 2. Model variables and sources.**

| Variable | Variable Code | Source |
|---|---|---|
| **Voice and Accountability** | | |
| Percentage of voters attending elections in each municipality | VA voter percentages | TSE |
| Number of mayoral candidates | VA number of candidates | TSE |
| Whether a female mayor had served in office | VA female mayor | TSE |
| Existence of a city hall internet page | VA webpage | PMB/IBGE |
| Number of companies in information and communication sectors | VA communication companies | CEMPRE/IBGE |
| **Government Effectiveness** | | |
| Number of administrative employees (direct and indirect) | GE employees | IBGE |
| Participation in the intermunicipal consortium for housing, health, and urban development | GE consortiums | PMB/IBGE |
| Existence of a master plan | GE masterplans | IBGE |
| **Regulatory Quality** | | |
| Number of companies in the agricultural sector | RQ ag. companies | CEMPRE/IBGE |
| Number of companies in non-agricultural sectors | RQ non-ag. companies | CEMPRE/IBGE |
| Number of employees in agricultural companies | RQ ag. employees | CEMPRE/IBGE |
| Number of employees in non-agricultural companies | RQ non-ag. employees | CEMPRE/IBGE |
| Incentives for enterprise existence | RQ enterprise incentives | IBGE |
| Restrictions for enterprise existence | RQ enterprise restrictions | IBGE |
| **Rule of Law** | | |
| Existence of zoning law | ROL zoning law | IBGE |
| Existence of division of land law | ROL division of land law | IBGE |
| Existence of urban improvement contribution law | ROL urban improvement law | IBGE |
| Existence of urban neighborhood impact law | ROL urban neighborhood law | IBGE |
| **Environmental Governance** | | |
| Existence of environmental agencies | EG environmental agency | PMB/IBGE |
| Number of employees in environmental agencies | EG environmental employees | IBGE |
| Existence of environmental municipal council | EG environmental council | PMB/IBGE |
| Existence of municipal environmental fund | EG environmental fund | PMB/IBGE |
| **Controls** | | |
| Population density (people/km$^2$) | Population density | IBGE |
| Crop density (crops/km$^2$) | Crop density | PAM/IBGE |
| Cattle density (cattle heads/km$^2$) | Cattle density | PPM/IBGE |
| Gross domestic product (per person) | GDP | IBGE |

TSE—The Superior Electoral Court, PMB/IBGE—Brazilian Municipalities Profile, CEMPRE/IBGE—Central Business Register, IBGE—The Brazilian Institute of Geography and Statistics, PAM/IBGE—Municipal Agricultural Production, PPM/IBGE—Municipal Livestock Profile.

three four-year periods (2005–2008, 2009–2012, and 2013–2016), which correspond to the mayors' election year mandate in Brazil. We used three election year cycles because this was the longest span of consistently available data at the time of our study's data collection. The variables we collected comprised a combination of continuous and categorical (presence/absence) data. In cases where we had multiple entries per time period, we calculated one value. For continuous variables (such as for annual data), we averaged the data in each time period. For categorical variables, we classified the entries into presences and absences, where any time period with at least one presence was classified as such. We normalized variables that likely correlated with population size (GE employees and RQ agricultural companies) by dividing them by the population of the municipality. We omitted all variables that were not available

for at least three time periods (e.g. those from the IBGE Census of Agriculture), had data collection or reporting processes that were inconsistent over time, did not vary over time, and were not spatially available across all study municipalities. The final dataset consisted of 22 variables that represent five governance indicators (Table 2). See the Supporting Information for more information on variable definitions (S1 Appendix).

**2.3.2 Dependent variable: Average yearly deforestation rate.** We used official data on annual deforestation for all municipalities in the Brazilian Amazon, which was sourced from Brazil's publicly available PRODES Project platform [53]. We defined average yearly deforestation rate as the total square kilometers of primary forest cover cleared over each time period divided by the number of years considered, which enabled us to calculate one deforestation rate for each of the three time periods. We additionally calculated the average yearly deforestation rate for a baseline period (2001–2004) and a fourth time period spanning the years 2017 and 2018 to allow for a lagged model specification. The deforestation data was strongly right-skewed and followed a log-normal distribution. Hence, we log-transformed the deforestation metric in all time periods to reduce the skew of the model residuals and improve symmetry.

**2.3.3 Control variables.** We selected a set of time-variant control variables in line with previous research [e.g. 72, 83, 84] to account for other direct and underlying drivers of deforestation [7]. These included cattle density, crop land density, population density, and gross domestic product. We did not estimate time-invariant controls such as density of protected areas and indigenous lands due to the fixed-effects model specification.

## 2.4 Phase three: Model specification

To evaluate the relationship between governance variables and deforestation, we specified a spatial panel fixed-effects regression model that related deforestation activity in each time period to municipal governance variables from the previous time period. This lagged model specification assumed that changes in local governance manifested over time periods longer than four years. We preferred this specification because it removed some endogeneity concerns between the explanatory variables and deforestation outcomes within the same time period. Formally, this model is specified in Eq (1):

$$y_{it} = \lambda y_{it-1} + X_{it-1}\beta + \alpha_i + \alpha_t + \varepsilon_{it} \tag{1}$$

$$\varepsilon_{it} = \rho W \varepsilon_{it} + v_{it} \tag{2}$$

for $i = 1, 2, \ldots, 457$ municipalities and $t = 3$ time periods, where $X$ is a matrix of independent variables in time period $t$ -1, $\beta$ is a vector of regression coefficients, $\lambda$ is the coefficient for a one time period lag of the dependent variable, $\alpha_i$ and $\alpha_t$ are vectors of unobserved individual and time effects, and $\varepsilon$ is an error term composed of spatially structured error (with spatial autocorrelation coefficient $\rho$ and neighborhood weights matrix $W$) and independently normally distributed error $v$ (Eq 2). We chose a spatial-error model structure after confirming the presence of spatially autocorrelated residuals in a standard fixed-effects panel regression (see S1 Text and S2 Fig).

Using our dataset of governance variables and controls, we ran several fixed-effect panel regressions using the *plm* [85] and *splm* [86] packages in R statistical software version 3.6.3 (R Core Team, 2019). We performed a series of robustness checks on alternate model specifications including a controls-only subset, a significant variables subset, a two-indicator subset, and an unlagged model (S3–S6 Tables).

## 3. Results

### 3.1 Deforestation dynamics in municipalities in the Brazilian Amazon

Average annual deforestation in the 457 study municipalities decreased from 2005 to 2018 (Fig 1). During the study period, the total deforested area was 115.4 thousand km$^2$, though rates of deforestation varied for each year within each time period. The largest drop in deforestation occurred between Period 1 and Period 2.

Deforestation also varied across space. Forest loss was concentrated along the frontier of deforestation—a swath of land located from East to West along the Southern rim of the basin (Fig 2). Along this frontier, deforestation primarily occurred in tandem with infrastructure development [87, 88], the expansion of agricultural commodities [33], illegal logging [89], population and urban growth, land grabbing and conflicts [90, 91], and weakening of federal environmental governance [92]. Out of the 457 municipalities, four were responsible for more than 15.90% of total deforestation during the study period. São Félix Do Xingu in the state of

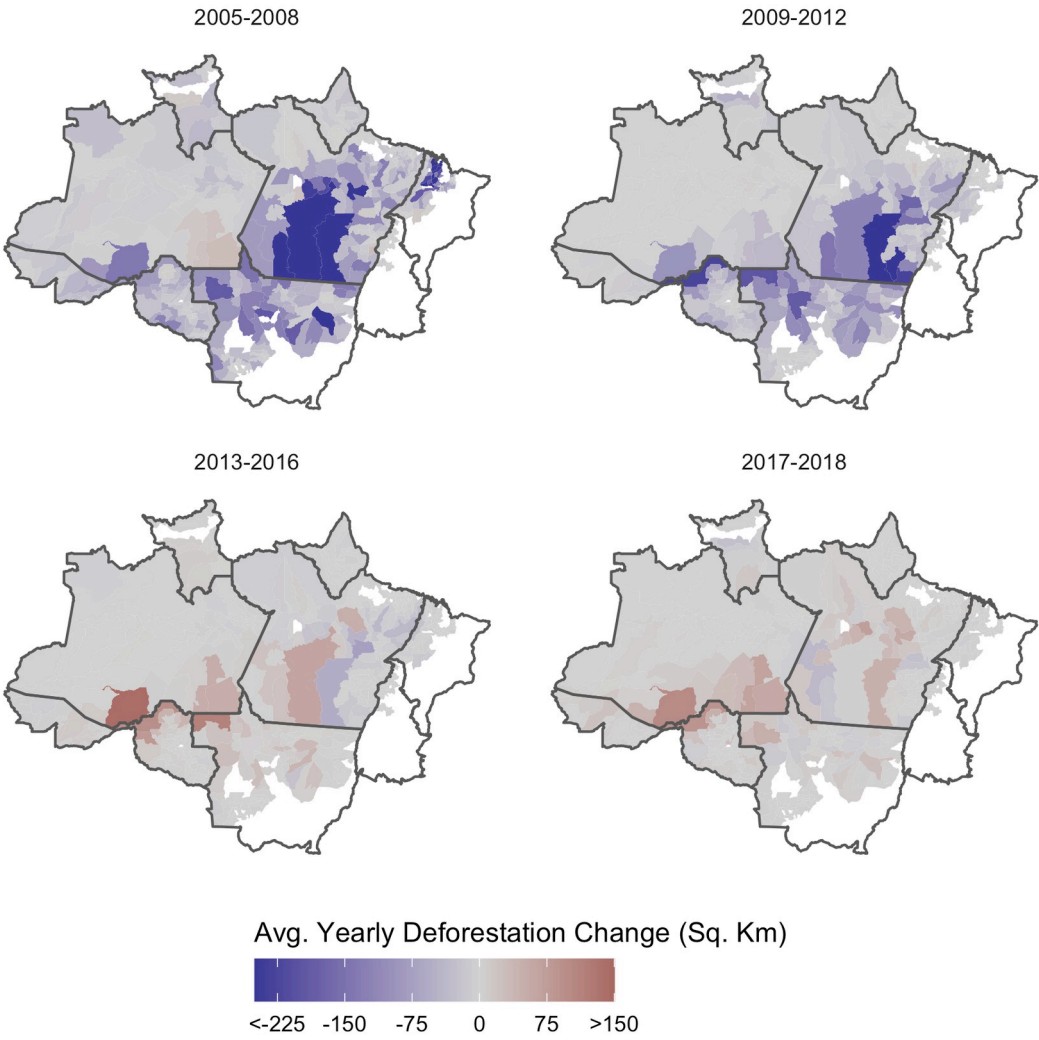

**Fig 2. Period-to-period changes in average yearly deforestation.** Red municipalities represent increased deforestation compared to the previous period, while blue municipalities represent decreased deforestation. Areas with the greatest amount of change represent the frontier of deforestation.

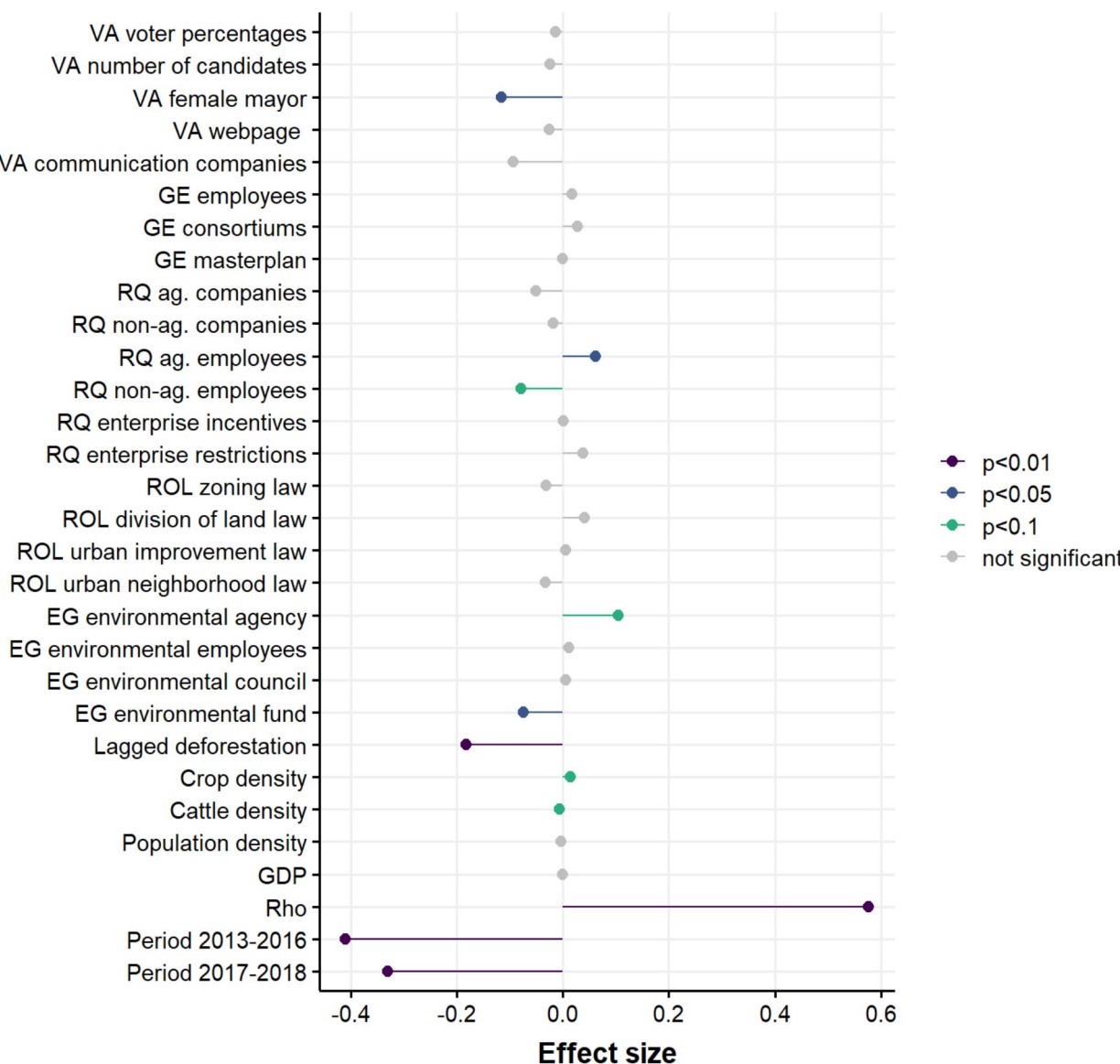

**Fig 3. Coefficient estimates of each variable in the lagged spatial panel regression model at three significance levels (p<0.01, p<0.05, and p<0.1).** The acronyms before each variable name represent the governance indicators of Environmental Governance (EG), Government Effectiveness (GE), Rule of Law (ROL), Regulatory Quality (RQ), and Voice and Accountability (VA). Lagged deforestation represents the log transformed deforestation rate from the t-1 time period. Rho corresponds to the spatial autocorrelation coefficient. Period 2013–2016 and Period 2017–2018 are time period fixed effects.

Pará ranked first (6.34 thousand km$^2$), followed by Altamira in Pará (4.56 thousand km$^2$), Porto Velho in Rondônia (4.31 thousand km$^2$), and Novo Repartimento in Pará (3.14 thousand km$^2$).

## 3.2 Primary relationships between governance variables and deforestation

Five of the 22 governance variables included in the model were significantly associated (p<0.01, p<0.05, and p<0.1) with municipal-level deforestation rates in the Brazilian Amazon between 2005 and 2018 (Fig 3). The presence of an environmental agency was associated with 10% higher rates of deforestation, the presence of an environmental fund was associated with

7% lower rates of deforestation, the number of employees working in agricultural companies was associated with 6% higher rates of deforestation, the number of employees working in non-agricultural companies was associated with 8% lower rates of deforestation, and the presence of a female mayor was associated with 12% lower rates of deforestation (S2 Table). The indicators of environmental governance and regulatory quality each had two variables associated with deforestation, although the variables representing regulatory quality may have been heavily influenced by the direct drivers of deforestation (see Discussion).

Our results also demonstrated significant relationships for the control variables of cattle and crop density (p<0.01) and highly significant relationships (p<0.001) for lagged deforestation, time period fixed effects, and the spatial autocorrelation coefficient (Rho). The effect sizes of the time period fixed effects and lagged deforestation were several magnitudes larger than the effect of any governance variable. These effect sizes likely corresponded to the large reduction in deforestation that occurred across the study region.

In the four alternate model specifications, we found that the coefficient values for all models were robust to different model variations, with the exception of the variable environmental agency, which was not significant in the alternative models (S3–S6 Tables).

### 3.3 Spatial patterns of governance variables

To help contextualize the model results, we visualized period-to-period changes for each of the significant governance variables (Fig 4). The positive link between higher deforestation rates and larger numbers of employees in agricultural companies was consistent with our prior expectations, since the largest increases consistently occurred along the frontier of deforestation, notably in the southern Amazon in the state of Mato Grosso, which was the largest producer of soy commodities in Brazil. The number of employees in non-agricultural companies increased in Mato Grosso and Pará, a trend that was also observed to some degree across the entire region. For both variables, the changes were relatively similar across both time periods, although more municipalities had decreases in the number of employees during the 2013–2016 period compared to 2009–2012. Changes in the existence of an environmental agency and environmental fund showed slightly different patterns, reflecting their opposite association with deforestation. The establishment of municipal environmental agencies, which was associated with higher deforestation rates, was most prevalent in the 2013–2016 time period and was concentrated in municipalities in the states of Amazonas, Acre, Pará, Roraima, and Rondônia. The spatial patterns for changes in the environmental fund was less clear, with municipalities implementing environmental funds in the southern, northern, and eastern portions of the Amazon in the 2009–2012 time period and across the entire region in 2013–2016. Both variables also demonstrated that a number of municipalities removed an environmental agency or fund only to reestablish it in a later period, indicating that environmental governance initiatives were sometimes impermanent over the mayors' election year mandate. The existence of a female mayor did not show a strong spatial trend in the study period, since some municipalities in every state elected women to the mayoral office.

## 4. Discussion

Amongst the wide range of factors that contributed to deforestation in the Brazilian Amazon, our results demonstrate that several variables related to local governance played a role in deforestation dynamics at the municipal level. In particular, this study identified that the variables of agricultural employees, non-agricultural employees, environmental fund, environmental agency, and female mayor were significantly related to deforestation. Changes in variables related to the indicators of environmental governance and regulatory quality were most closely

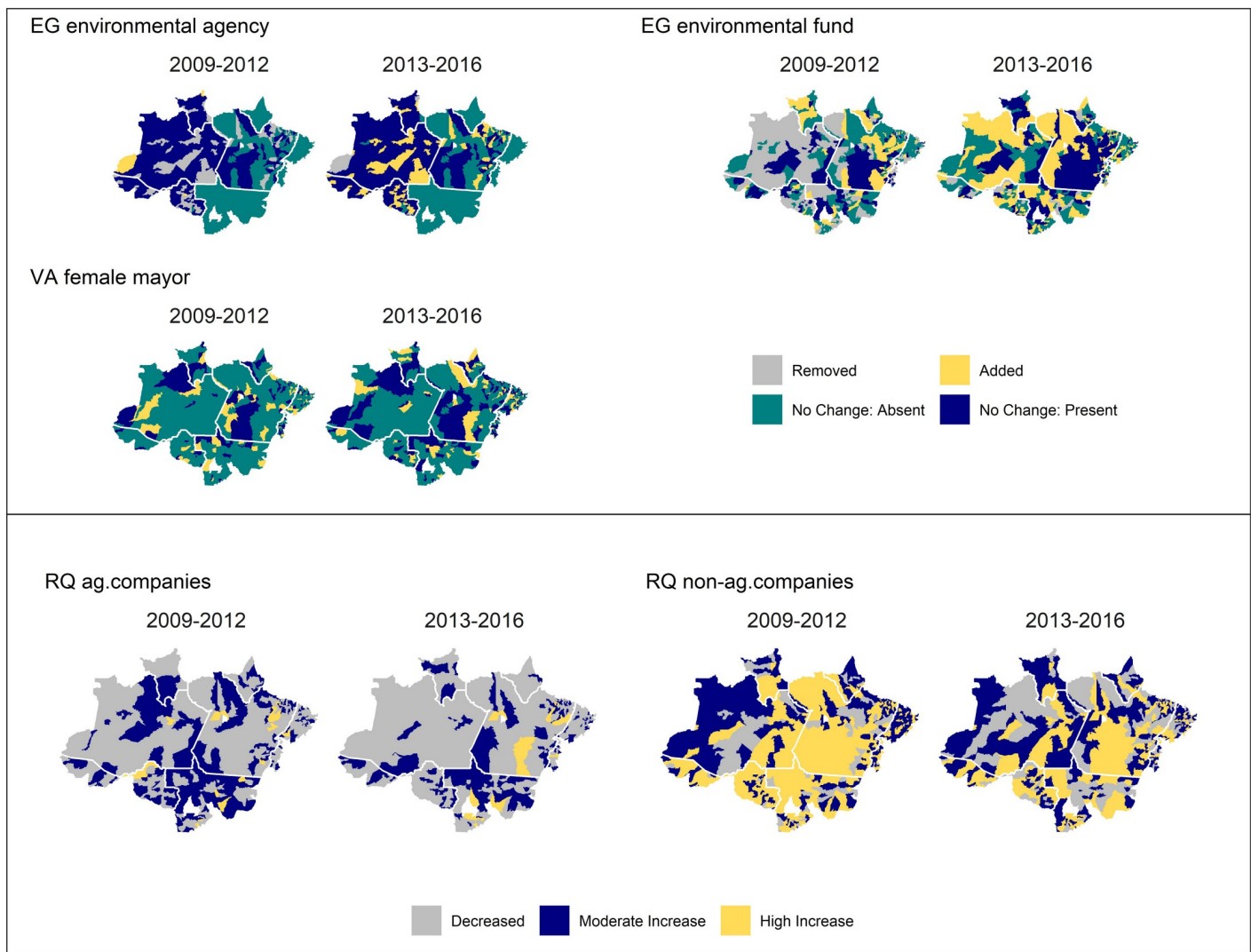

**Fig 4. Period-to-period changes in significant variables in the model.** Positive changes greater than 100 units were classified as a high increase and positive changes between 0–100 were classified as a moderate increase.

associated with changes in deforestation across time periods. However, the variables representing regulatory quality may have also captured variation for the effects of more direct drivers of deforestation (e.g. agricultural expansion). Spatially, the changes in the two variables that represented regulatory quality were most pronounced along the frontier of deforestation such as in Mato Grosso, while the greatest changes in the two variables that represented environmental governance were most pronounced in the northwestern states such as Amazonas. While our findings revealed the potential for specific local governance attributes in mediating and combating deforestation, our study also suggests that subnational governance alone will not be sufficient to tackle the complexity of forest loss. Specifically, the spatial autocorrelation and lagged deforestation model specification indicated that broader-scale processes and external factors driving the expanding deforestation frontier were large contributors to deforestation trends. Similarly, broader patterns of deforestation were also likely to have been influenced by federal and state government interventions to reduce deforestation across the biome (e.g. PPCDAm) [74]. Our study therefore builds upon knowledge that both the direct drivers of

deforestation and underlying drivers such as local governance contribute to deforestation [7, 15, 16].

Below, we focus on the significant variables in the model, discussing their associations with deforestation and the potential implications of our findings for municipal-level governance in the Brazilian Amazon. We then discuss how our results differed from our expectations for the variables that did not have a significant association with deforestation. Finally, we reassess the governance framework used in this study and identify directions for future research.

## 4.1 Expansion of the agricultural sector related to deforestation

We found that higher deforestation rates were associated with increasing numbers of employees in agricultural companies. These variables may have been dually linked to agricultural expansion–a direct driver of deforestation–as well as the underlying driver of regulatory quality. As such, in this section we discuss the significance of agricultural expansion in terms of both the direct driver and the governance indicator of regulatory quality. In terms of agricultural expansion, increasing numbers of employees directly translates to agribusinesses having a greater ability to deforest. In terms of regulatory quality, our findings correspond to another study that suggested that regulatory quality facilitated deforestation [45] and contributes to a body of literature with mixed evidence on the direction of this relationship [46]. The relationship between expansion of the agricultural sector and private sector development is important at the municipal level since municipal-level governance may either promote or regulate agricultural expansion, either boosting or decreasing the amount of deforestation that occurs. The link between agricultural employees and deforestation is particularly relevant across the Brazilian Amazon. The expansion of companies and jobs in the agricultural sector is predominantly associated with cattle ranching and annual crop production (including soy), which are major agricultural activities driving deforestation in the Amazon [93–97]. Specifically, these activities were concentrated in the states of Pará, Mato Grosso, Maranhão, and Rondônia, where annual crops expanded over pasturelands, driven by international market demands that pushed cattle ranching to the fringes of the frontier. Some municipalities have lessened the impact of this agricultural expansion in their territories by partnering with international conservation NGOs [64]. Others may have promoted agricultural expansion by loosening environmental monitoring or by participating more often in federal credit programs that increased incentives driving forest loss [98]. In other cases, these results may have corresponded to a broader trend where powerful interests, such as wealthy elites or large agribusinesses, gained control over municipal governments and diverted power from the state [99]. This phenomenon of elite capture has often enabled farmers, land speculators, agribusiness enterprises, and ranchers to more easily expand their businesses while loosening or restraining forest regulations [12, 61, 100].

Our results also show that lower deforestation rates were associated with increasing numbers of employees in non-agricultural companies, although this result had less support since it was marginally significant (p<0.1). One explanation for this result is that municipalities that were already deforested through previous boom-and-bust cycles of agricultural frontier expansion may have had enough available land and resources to transition and expand into additional industries, hiring more employees in economic sectors external to agriculture. The number of non-agricultural employees may have also increased along with urban growth [101], to a greater extent than what was captured by the population density control variable. It is also possible that municipalities with forest-based economies were associated with lower deforestation because local forest-based livelihoods have incentivized conservation [102]. To clarify this relationship, future research may consider exploring how variables such as available

land, urbanization, and forest-based livelihoods have influenced the ability of governments to regulate deforestation driven by agricultural expansion. One such analysis would become possible with datasets that can more clearly distinguish between the effects of the direct agricultural drivers from the effects of the underlying governance drivers.

## 4.2 Environmental fund related to less deforestation while environmental agencies related to more deforestation

We found that lower deforestation rates were linked to the creation and implementation of the municipal environmental fund. These results support findings from past studies, which demonstrated that efforts specifically targeted at improving environmental governance contributed to preventing deforestation, and correspond to our expectations [34, 43, 51]. In Brazil, the municipal environmental fund seeks to collect and provide local government officials with resources (e.g. from environmental fines and licensing fees or green taxes) to support and advance local environmental projects and programs. Previous studies of incentive-based funding programs aimed at reducing deforestation in the Amazon found that they were often effective [103] and promoted local land tenure security [104]. Yet there is still debate surrounding which environmental programs should be funded, how they should be funded [105, 106], and who should be providing the funds [107]. While some scholars argue that large upfront investments are necessary to catalyze positive change [106, 108], others argue that upfront investments are wasted if investments in local capacity are ignored [105, 107]. Regardless, the association between the environmental fund and deforestation demonstrates the importance of funding or incentives to combat deforestation and/or promote sustainable initiatives and livelihoods. Given that the municipal environmental fund requires the design and approval of bills through a management board, the existence of a fund may indicate local government officials' commitment to collaboratively address environmental degradation and work to improve environmental governance.

Contrary to our hypotheses, our results suggest a positive association between deforestation rates and the implementation of municipal environmental agencies, although this result had slightly less significance (p<0.1) and was not significant in the other model specifications (S3–S6 Tables). This finding contradicted our expectations as previous studies found a relationship between decreased deforestation and the presence of environmentally focused stakeholders including NGOs [34], extension agents [109], and environmental observers [110]. This finding could be partially explained by national and state efforts to decentralize environmental governance programs to the municipal level. Such programs resulted in investments in hiring, training, and capacity building of environmental agents in municipal secretariats. It is also possible that state-led programs specifically targeted the implementation of environmental agencies to those municipalities with the most deforestation. For example, since Brazil's LPM policy initiated municipal-level environmental action to target key deforestation hotspots, the creation of environmental agencies may have focused on areas that were already experiencing high deforestation rates. Alternatively, our results may represent increased decentralization that was not followed by improved quality or effectiveness of environmental agencies. This raises questions on both the potential and limits of municipal environmental governance. While some studies have suggested that strong local governance can make up for weaker, or absent, governance at higher levels [111], others have emphasized the importance of comprehensive federal governance [112, 113]. In addition, by demonstrating that governance indicators may not always have the expected relationship with environmental outcomes, this result emphasizes the importance of considering context-specific governance dynamics that may influence theoretical relationships.

One additional area of future research is to investigate whether this trend indicates a reactive rather than proactive approach to environmental protection. If it was the case that environmental initiatives were more reactive to deforestation, then municipalities that experienced higher deforestation rates may have responded by hiring additional environmental employees to address the problem. This finding may highlight the need for more anticipatory approaches to reduce deforestation.

## 4.3 Female leadership related to reduced deforestation

We found that electing a female mayor was associated with lower rates of deforestation. This supports findings from other studies linking women's leadership in governance with positive environmental outcomes. For instance, corporate firms with women serving on the board of directors have been more likely to implement corporate social responsibility practices [114], and community forests with women serving on the executive committee have had better forest conservation outcomes [115]. Although we classified female mayors as representing the voice and accountability governance indicator [116], it is possible that female leadership also represents other indicators, including social equity [117] and control of corruption [118]. This finding may suggest that women leaders contributed to reducing deforestation, or that municipalities that elected women leaders also had more successful environmental programs to reduce deforestation.

## 4.4 Several variables did not relate to deforestation

We expected that the variables representing government effectiveness would correlate with lower rates of deforestation. We anticipated this link given that several studies found a positive correlation between government effectiveness and improved development/citizen well-being [119, 120] and between well-being and conservation outcomes [121–123]. While government effectiveness may have resulted in improvements in access to basic services and citizen well-being, it is possible that these improvements were not sufficient or did not correlate with deforestation. For example, governments may not have promoted local enforcement of federal conservation initiatives, opportunities for sustainable supply chains, or incentives for forest conservation. It is also possible that the variables used to represent government effectiveness were not ideal representations of the concept and that additional data may reveal different trends.

We expected to find a negative relationship between the variables representing rule of law and deforestation. We anticipated that as rule of law increased, deforestation would decrease due to improving enforcement of conservation policies. Past studies found either a positive association between rule of law and reduced deforestation [34] or no association [124]. This variability highlights the need to further investigate the relationship between rule of law and deforestation. In Brazil, there is an important difference between the existence of policies aiming to reduce deforestation and the enforcement of these policies. While Brazil is considered to have one of the strictest environmental law systems in the world, it faces enormous challenges with enforcement [125, 126]. Data availability is a challenge in highlighting this important nuance: while data on the existence of environmental laws at the municipal level is readily available, the quality of enforcement at this level is more difficult to measure. Rather than measuring the existence of environmental laws, government agencies may consider sharing metrics related to law enforcement outcomes, such as arrests made and successful prosecutions. Given that another study [33] similarly attempted to measure municipal-level rule of law in the Brazilian Amazon using territorial planning laws, but also found no significant effect, it

may be worthwhile for government agencies to consider collecting and sharing data that more directly assess outcomes of effective rule of law.

The expected relationships between the variables representing voice and accountability and deforestation are not entirely clear since scholars have found both positive [35] and negative associations [34, 38, 41]. The direction of this relationship may depend on the local population's perspective on forest conservation. For example, in 2019 a number of farmers in the municipality of Altamira set fires to visibly support anti-environmental policies promoted by the federal administration [127]. Conversely, indigenous leaders in the Amazon have often been murdered while fighting to protect forested land [128]. These examples raise concerns about who speaks and when, especially given perceived tensions in the region between conservation and development [94, 129].

## 4.5 Reflections and recommendations for improving the governance framework

Our study indicates that the relationship between governance and deforestation at the municipal level in the Brazilian Amazon is important for certain variables but not for others. The modified WGI framework used in this analysis enabled us to better understand which variables contributed to the concept of governance and our methodology provided a template for how publicly available datasets can be used to analyze governance at the municipal level.

Our study also highlights several ways that studies utilizing the WGI framework may be modified to better address municipal-level forest or environmental governance. Specifically, the fact that certain governance variables contributed to increased deforestation reflects a critique voiced by scholars that the WGI framework puts too much emphasis on economic means of measuring well-being, by including the successes of businesses as one of the primary governance indicators [130]. This emphasis may prioritize the interests of business elites and/ or local government revenues over environmental protection [130, 131]. Similar to other studies, our research supports the inclusion of an environmental governance indicator when analyzing deforestation trends [33, 34, 43, 51, 81].

We furthermore observed that concepts such as social equity have not been included in many governance frameworks. Research has shown that economic inequities have exacerbated forest degradation, while collective action institutions that reduce social inequities have improved forest management [132]. Decentralized natural resource governance does not automatically correct for power imbalances or the inequitable distribution of benefits within a community, especially when demographic factors such as gender, indigeneity, religion, poverty, or residency status limit eligibility criteria for decision making [133–135]. Including an indicator that captures social equity may allow researchers to determine how deforestation varies according to the power dynamics of actors, their positions, and their degrees of access to resources and information.

## 4.6 Methodological considerations

The primary limitation of our analysis was data availability and quality. As a result, for some of the governance indicators, the combination of variables were imperfect representations of the selected indicators. There were several reasons for this. First, much of the available data was originally collected for other broader purposes, such as for economic, social, and demographic statistics, and therefore did not translate to ideal proxies for governance indicators. Second, we removed two indicators from the original WGI framework because the data sources for these indicators were not continuous across our analytical timeframe and therefore could not be included in the study. Governance would have been better represented by including data on

control of corruption and political stability. Third, there was a general lack of data availability for key measures we had hoped to track at our desired temporal and spatial scales. Some metrics that were not available could have feasibly been measured across municipalities and shared publicly, for example, those pertaining to enforcement of federal laws at the local level, though we were unable to locate any such metrics.

## 4.7 Directions for future research and policy implications

Our study aimed to fill a gap in understanding municipal-level forest governance. Investigations that integrate socio-environmental data across understudied levels and scales can reveal important relationships that have implications for land use policy and conservation outcomes. Since local authorities and actors may influence politics at regional and global levels, it is useful to conceptualize the role of forest governance at local levels in addition to the more commonly studied aggregated levels of regional and national scales [26, 136, 137]. Forest governance is a multi-actor, multi-sector, and multi-level system and improving initiatives at local levels may also contribute to improvements at more intermediate levels [14, 26]. Measuring governance at local levels is therefore important and can assist in both understanding changes in deforestation and in communicating the role of local-level governance to policymakers. Yet few studies have sought to systematically address questions across subnational scales. This study demonstrated that it is possible to synthesize local-level governance and deforestation data. However, this novel aspect of our approach also created challenges due to a lack of data availability.

In future research, we recommend investigating at which time scale different governance indicators and processes occur. While deforestation occurs on a short time scale and is visually measurable, governance and other drivers of deforestation occur across a longer timescale and are difficult to measure. To more precisely analyze changes in governance indicators, more research is needed on the time scales at which it is possible to measure changes in governance. For example, it could be the case that regulatory quality and environmental governance change significantly over the time period from 2005 to 2016, while other indicators, such as government effectiveness or rule of law develop more gradually over time. In addition, since informal rules and norms are also important contributors to municipal-level forest governance, research on strengthening informal governance structures and boosting environmental funding for these structures, including for community-level leadership, social movements, or civil society, may expand knowledge on the range of governance initiatives that reduce deforestation.

This study highlights that more original data is needed on the role of municipal-level governance that is consistent across time and space. Given the need to better identify trends and causal relationships between governance and deforestation, we encourage future studies to engage with available data despite existing limitations. Future research that relies on interviews with local stakeholders in a cross-section of municipalities in the Brazilian Amazon may shed further light on the relationships highlighted in this paper, as the analysis of both publicly available data and perceptions data will be important to understand the role of governance on deforestation. Studies that work with representatives from Brazilian municipalities to develop place-based metrics for understanding forest governance will advance understanding of municipal-level forest governance while identifying better local-level indicators for monitoring and evaluating forest governance. This will enable future studies to provide a clearer picture of the effectiveness of governance on the ground [82]. Qualitative data collection and subnational-level perceptions of governance data would strengthen broader understanding of variations in local-level governance. Future research should build upon this study by continuing to integrate publicly available socio-environmental data sources to uncover the complex mix of factors that drive land use changes across the globe.

## 5. Conclusions

Our research found indications that municipal-level governance matters for deforestation in the Brazilian Amazon, with implications for subnational governance in other countries with multilevel forest governance systems. We found that the variables that represented existence of an environmental fund, non-agricultural employees, and female mayors had negative relationships with deforestation, while the variables that represented number of agricultural companies and implementation of an environmental agency had positive relationships with deforestation. These results suggest that governance at the municipal level does not uniformly relate to reduced deforestation. Rather, different variables and indicators of governance may individually relate to either increased or decreased deforestation. We expect that future studies that leverage data sources specifically designed for governance assessments, rather than publicly available data sources, may find even stronger relationships between governance and deforestation.

The variable that we believed was most relevant to providing recommendations to policy- and decision-makers was the relationship between environmental fund and deforestation. One direction for future research is to investigate the causal direction of this relationship. If the existence of an environmental fund has been able to effectively reduce deforestation, then increased municipal environmental funding and/or more frequently institutionalized municipal environmental governance could lead to further reductions in deforestation. Additionally, if high deforestation rates have caused municipalities to increase the numbers of environmental government agencies, employees, etc., but these structures have not been able to effectively reduce deforestation rates, then improvements of municipal environmental governance structures may benefit environmental goals.

To suggest actionable outcomes for municipal-level decision-makers, more research is needed on the specific conditions that allow for stronger environmental governance to influence deforestation rates. Understanding why forests are better conserved through local governance in certain localities and not others would allow decision-makers to tailor policy according to local-level drivers of deforestation, to both improve municipal environmental governance and to protect forests. By synthesizing governance theory and econometric modeling, this study was an important step in analyzing the relationship between municipal-level governance and deforestation.

## Supporting information

**S1 Appendix. Glossary.**
(DOCX)

**S1 Fig. Model fit and residual plots for the lagged spatial panel regressions including all governance variables.**
(DOCX)

**S2 Fig. Maps of the model residuals for the non-spatial and spatial lagged panel regressions.** Municipalities are colored based on the model residuals (difference between fitted and observed values) by time period.
(DOCX)

**S1 Text. Spatial autocorrelation test.**
(DOCX)

**S2 Text. Alternate model specifications.**
(DOCX)

**S3 Text. Model comparison.**
(DOCX)

**S1 Table. All data sources reviewed for the study.** Only some of the reviewed sources were used in the final dataset.
(DOCX)

**S2 Table. Model parameters for the all governance variables model with a lagged model specification.**
(DOCX)

**S3 Table. Model parameters for the controls only model with a lagged model specification.**
(DOCX)

**S4 Table. Model parameters for the significant variables only model with a lagged model specification.**
(DOCX)

**S5 Table. Model parameters for environmental governance and regulatory quality variables with a lagged model specification.**
(DOCX)

**S6 Table. Model parameters for the all governance variables model with an unlagged model specification.**
(DOCX)

**S7 Table. Akaike information criterion across model specifications and predictor subsets.**
(DOCX)

## Acknowledgments

We acknowledge support from Dr. Nicole Motzer, Dr. Jonathan Kramer, Dr. Peter Richards, and Dr. Peter Newton. We are indebted to GoverNancy for offering us both carrots and sticks towards the completion of this research.

## Author Contributions

**Conceptualization:** Rayna Benzeev, Bradley Wilson, Megan Butler, Paulo Massoca, Karuna Paudel, Lauren Redmore, Lucía Zarbá.

**Data curation:** Rayna Benzeev, Bradley Wilson, Megan Butler, Paulo Massoca, Karuna Paudel, Lauren Redmore, Lucía Zarbá.

**Formal analysis:** Rayna Benzeev, Bradley Wilson, Karuna Paudel, Lucía Zarbá.

**Funding acquisition:** Rayna Benzeev, Bradley Wilson, Megan Butler, Karuna Paudel, Lauren Redmore, Lucía Zarbá.

**Investigation:** Rayna Benzeev, Bradley Wilson, Megan Butler, Paulo Massoca, Karuna Paudel, Lauren Redmore, Lucía Zarbá.

**Methodology:** Rayna Benzeev, Bradley Wilson, Megan Butler, Paulo Massoca, Karuna Paudel, Lauren Redmore, Lucía Zarbá.

**Project administration:** Rayna Benzeev, Bradley Wilson.

**Visualization:** Rayna Benzeev, Bradley Wilson, Karuna Paudel, Lucía Zarbá.

**Writing – original draft:** Rayna Benzeev, Bradley Wilson, Megan Butler, Paulo Massoca, Lauren Redmore, Lucía Zarbá.

**Writing – review & editing:** Rayna Benzeev, Bradley Wilson, Megan Butler, Paulo Massoca, Karuna Paudel, Lauren Redmore, Lucía Zarbá.

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
