## [Decision Letter · Decision Letter 0]

2 Feb 2022

PONE-D-21-31445What’s governance got to do with it? Examining the relationship between governance and deforestation in the Brazilian AmazonPLOS ONE

Dear Dr. Benzeev,

Thank you for submitting your manuscript to PLOS ONE. After careful consideration, we feel that it has merit but does not fully meet PLOS ONE’s publication criteria as it currently stands. Therefore, we invite you to submit a revised version of the manuscript that addresses the points raised during the review process.

Reviewer 1 and Reviewer 2 have very different overall recommendations for you to consider, but in general both are asking for more context for your results. Reviewer 2 is more critical, and therefore I would expect that addressing their comments may take much more effort, but have some significant potential to enhance the audience of this paper. In sum, thorough consideration of reviewer 2's comments is likely to significantly improve the paper. I would note that their indications that the concept of "governance" should be more thoroughly elaborated, and some of their discussion of data and definitions were particularly resonant for me when I read the paper (both reviewer 2, whom I do not know personally, and myself are knowledgeable about deforestation in Amazonia and econometric approaches to model this deforestation). While I do not necessarily consider the revision a full-on "major revision" and do agree with reviewer 1's generally positive assessment, I believe that some of the issues reviewer 2 raises push the revision to something in between "minor" and "major."

We look forward to receiving your revised manuscript.

Kind regards,

Stephen P. Aldrich, PhD

Academic Editor

PLOS ONE

Journal Requirements:

2. We note that Figure 1, 2 and 4 in your submission contain [map/satellite] images which may be copyrighted. All PLOS content is published under the Creative Commons Attribution License (CC BY 4.0), which means that the manuscript, images, and Supporting Information files will be freely available online, and any third party is permitted to access, download, copy, distribute, and use these materials in any way, even commercially, with proper attribution. For these reasons, we cannot publish previously copyrighted maps or satellite images created using proprietary data, such as Google software (Google Maps, Street View, and Earth). For more information, see our copyright guidelines: http://journals.plos.org/plosone/s/licenses-and-copyright.

1. You may seek permission from the original copyright holder of Figure 1, 2 and 4 to publish the content specifically under the CC BY 4.0 license.  

Reviewers' comments:

Reviewer's Responses to Questions

**Comments to the Author**

1. Is the manuscript technically sound, and do the data support the conclusions?

Reviewer #1: Yes

Reviewer #2: Yes

2. Has the statistical analysis been performed appropriately and rigorously? 

Reviewer #1: Yes

Reviewer #2: I Don't Know

3. Have the authors made all data underlying the findings in their manuscript fully available?

Reviewer #1: Yes

Reviewer #2: Yes

4. Is the manuscript presented in an intelligible fashion and written in standard English?

Reviewer #1: Yes

Reviewer #2: Yes

5. Review Comments to the Author

Reviewer #1: The manuscript brings a great contribution to the literature gap regarding local government governance and its relation to deforestation in the Brazilian Amazon. Besides this main contribution, I would highlight the effort done to adapt governance indicators based on available data on Brazilian municipalities. The econometric approach is, up to my knowledge, robust, and the biases are addressed and disclosed properly. Also, the discussion presented on the results is enriching and demonstrates knowledge of the pertinent related literature. My recommendation is to accept the manuscript.

That said, I have some specific comments and questions for the authors regarding the manuscript, all of which I will state below.

(Comment 1) : One of the main contributions of the manuscript is, in my opinion, the process used by the authors to develop a set of indicators to evaluate the Brazilian municipal government, given the specific data available on municipalities in the country. This can be replicated, adapted, or expanded in future studies on municipal governance in Brazil – even if the focus is not related to deforestation.

(Comment 2) : The effect sizes of the time period fixed effects and lagged deforestation being several magnitudes larger than the effect of any (local) governance variable can indicate the possibility that the reduction of deforestation in the 1st and 2nd period analyzed is mainly related to supra-municipal government interventions (most probably federal government policies at the time), such as the PPCDAm.

(Comment 3) : Another finding I think is very interesting is the positive association between deforestation rates and the implementation of municipal environmental agencies in the lagged model. Although it can be an effect of command and control policies (such as LPM) targeting municipalities with the highest deforestation rate, as already pointed in the manuscript. This result serves to warn local environmental governance studies to context-specific dynamics that may prove counter-intuitive. In my opinion, the authors could stress this a little more.

(Question 1) : The authors surveyed different public databases but did not mention the IBGE Agricultural Census (Censo Agropecuário), in which there is information for the municipal level on land tenure, number of properties, and their sizes. Is there a particular reason why the authors have not included this in their model? In my opinion, one of the measures of local governance is related to the security of tenure and this is not present in the model.

Reviewer #2: The article presents results of a spatial panel fixed-effects regression model which relates publicly available governance and context variables from the Brazilian Amazon to deforestation in order examine relations between governance and deforestation at the municipal level. The article presents original new research, is overall well written and has a clear structure and line of argumentation that can logically be followed with conclusions that are based on the research findings.

The main draw back is a rather narrow perspective of the authors. They only have the Worldbank governance framework in mind and “their” municipality data. The study deserves and needs to be put into broader context. But this can be amended. A number of more specific issues are listed below. I recommend to publish this manuscript in PLOS One with some modifications and amendments.

50 – 56 The article would benefit from a clear governance definition, you mention loosely actors and practices, but do not provide a definition. There are so many on the market. You have a focus on mostly governmental plus some economic issues only which is rather narrow in view of a modern governance understanding. Mention, explain and discuss this. Also as concerns good governance – there are many concepts. Intro in general: how is governance related to other deforestation drivers? Provide a conceptual understanding that will at the end help considerably to interpret your findings – some more details are given below.

67 There is research on subnational levels, see Nansikombi et al (Forest Policy and Economics 120 (2020) and Fischer et al (World Development 148 (2021)

94-96 “Other studies have found that some governance indicators were correlated with negative outcomes for forests and the environment. For example, increases in indicators related to business and the economy“ – business and economy are not governance! e.g. Ceddia 2014 analyzes agriculture AND governance and their relations, but agriculture indicators are not governance.

129 “where the official monitoring system of Brazil (PRODES) detected deforestation during 2005-2018 “ better write “for which deforestation data were available”, otherwise it reads as if you excluded municipalities with zero deforestation, which you probably did not do, right?

168 Section 2.1.2 contains your results, all this is based on your evaluations and I recommend to include this as the first (still descriptive) subsection of you results.

197 there are many more , e.g. ….

Kishor, N., Kenneth, R., 2012. Assessing and Monitoring Forest Governance:: A user’s guide to a diagnostic tool. In, Program on Forests. PROFOR, Washington, D.C., USA.

Davis, C., Williams, L., Lupberger, S., Daviet, F., 2013. Assessing Forest Governance: The Governance of Forests Initiative Indicator Framework. In. WRI, Washington, D.C., USA.

de Graaf, M., Buck, L., Shames, S., Zagt, R., 2017. Assessing Landscape Governance, A Participatory Approach. In. Tropenbos, EcoAgriculture, Wageningen, Washington.

207 delete “perceptions of”

237 this is really a main constraint. It also becomes clear that you more or less pick what is available and from this pragmatic end, but not from a scientific perspective you design your study. E.g. Kaufmann on whom you base your concept says “Of the 31 data sources used in 2009, 5 are from commercial business information providers; surveys and NGOs contribute 9 sources each; and the remaining 8 sources are from public sector providers.” It is well discussed later in your paper but make this clear on a prominent place (title or abstract and conclusions), talk about “selected”, or “government perspective” … or. You argue that you do not rely on perceptions but on measured data – ok, but the drawback is that you need to take what is there, interpret this to make it fit in your categories, instead of asking/assessing the hard and essential governance factors.

258 there is no variable definition in the Supplementary, but would be interesting. E.g. “crop density” – you leave me alone with “(crops/km22) – PAM/IBGE “ what is this? At least two sentences in the supplementary to make sure the reader knows what is behind each of the indicators’ data, some info is there in the “Glossary”.

269-271 what is “original forest”, primary forest, secondary forest, any forest ? I do not find deforestation data on INPE 2020. Do you calculate the data yourself? How? based on deforestation maps, based on satellite data? This is the target variable so it deserves and understandable and complete description . “approximately conform to normality” does the model require normal distribution or not? Do your data fulfill the requirements, or not – how do you test this?

352 and following discussion

The discussion would benefit from a theoretical framework of how governance and other drivers are linked to deforestation. You already have Geist and Lambin in your reference list. Consider to introduce this as a framework in the Introduction. If you follow their idea of proximate/direct drivers and underlying causes, then you very obviously confirm this with your study; your context factors crops and cattle are the direct causes with “several magnitudes” stronger effects. Governance is underlying and thus much more complicated to show effects, also see Nanasikombi (2020) and Fischer (2021). If you apply this framework then it becomes clear that your indicators RQ ag. Companies, RQ non-ag. Companies, RQ ag. Employees, RQ non- ag. Employees must predominately be interpreted as direct driver indicators – not governance, as they mostly reflect the agricultural production in the area (even though they may as well reflect some regulative quality). In this sense I would be very cautious to claim that the two employee indicators (specifically with rather low p values) are a basis to claim that “local governance played a role in deforestation dynamics”. If you do not take them into account than you have environmental fund (negative), environmental agency (positive), and female mayor (negative) as remaining evidence (all with p<0.05 only) and I would interpret this more cautiously.

363 of course not silver bullet, direct drivers need to be tackled, but in all such measures governance may play a role – thus indirect driver, see above.

375 be much more cautious, see above

381 – 395 You see: now you are discussing the direct driver agriculture, not governance!

396 – 407 and again: you are not discussing regulatory quality but the direct drivers – even though you try to link it to regulatory quality in the last sentence which is a bit artificial.

471 – 473 yes

474 – 489 – nicely written and I agree

499 – 523 When discussing improvements in the Governance Framework you should show that you are aware of other frameworks, I gave some , see above. Then discuss why did you select this one? Others are designed completely different. There are many issues that are missing in the Worldbank framework compared to others.

514/515 I do not understand what you want to say

538/539 -skip this because you did not show anything about national data availability.

540-541 this could better be a subchapter on “methodological considerations” or alike, it has not so much to do with further research.

562 – 564 above you advocated that you use measured instead of perceived data, now you ask for perceived (interview) data. Perhaps both needed?

559 – 571 this is not only “future research” it has a lot of policy implications as well: you recommend to revise/amend public data collection/reporting, find other title.

574 – 576 this is of course true, I would nevertheless formulate more cautiously something like “found indications that m l governance matters” … and rather at the beginning mention the data base limitations by only using publicly available data that was mostly not designed for governance assessments, and: stronger statistical relations might be expected if the data could be improved.

590 – 593 did you research on informal rules? Which indicator was this? If not, then you should not conclude on this. Rather this is another indicator that may be missing in the World bank framework and could be mentioned in the discussion on amending the framework

In general I am pretty sure that you are not the first one to study municipal level governance – here is the result of 10 min lit search, also use “multilevel governance” and “landscape level governance” search terms

Municipal environmental governance in the Peruvian Amazon: A case study in local matters of (in)significance; P. B. Larsen;

Management of Environmental Quality 2011 Vol. 22 Issue 3 Pages 374-385

Secco et al. Forest Policy and Economics 49 (2014) 57–71

Scale and context dependency of deforestation drivers: Insights from spatial econometrics in the tropics, R. Ferrer Velasco, M. Kothke, M. Lippe and S. Gunter

PLoS One 2020 Vol. 15 Issue 1 Pages e0226830

Multilevel governance for forests and climate change: Learning from Southern Mexico

S. Rantala, R. Hajjar and M. Skutsch

Forests 2014 Vol. 5 Issue 12 Pages 3147-3168

Mixing carrots and sticks to conserve forests in the Brazilian amazon: A spatial probabilistic modeling approach

J. Börner, E. Marinho and S. Wunder

PLoS ONE 2015 Vol. 10 Issue 2

6. PLOS authors have the option to publish the peer review history of their article (what does this mean?). If published, this will include your full peer review and any attached files.

Reviewer #1: **Yes: **Vitor Bukvar Fernandes

Reviewer #2: No

---

## [Author Response · Author response to Decision Letter 0]

30 Mar 2022

Response to reviewers 

We thank the efforts of the Editor and two Reviewers for their careful consideration of our manuscript. We have seriously engaged with the comments from the reviewers and incorporated their changes. We are very grateful for the thoughtful suggestions. Our responses to each comment are shown in bold below (please see attached document for version with bold comments).

Reviewer #1: The manuscript brings a great contribution to the literature gap regarding local government governance and its relation to deforestation in the Brazilian Amazon. Besides this main contribution, I would highlight the effort done to adapt governance indicators based on available data on Brazilian municipalities. The econometric approach is, up to my knowledge, robust, and the biases are addressed and disclosed properly. Also, the discussion presented on the results is enriching and demonstrates knowledge of the pertinent related literature. My recommendation is to accept the manuscript.

That said, I have some specific comments and questions for the authors regarding the manuscript, all of which I will state below.

Thank you for your review. We are pleased that you highlighted that our article provides a contribution to a literature gap, utilizes available data, employs a robust approach, discloses biases, and demonstrates a knowledge of the literature in the discussion. We appreciate your thoughtful comments and suggestions, which significantly strengthened our paper. We respond to each comment below.

(Comment 1) : One of the main contributions of the manuscript is, in my opinion, the process used by the authors to develop a set of indicators to evaluate the Brazilian municipal government, given the specific data available on municipalities in the country. This can be replicated, adapted, or expanded in future studies on municipal governance in Brazil – even if the focus is not related to deforestation.

Thank you for your comment. We also hope that future studies will replicate, adapt, and/or expand the approach used in this manuscript. 

(Comment 2) : The effect sizes of the time period fixed effects and lagged deforestation being several magnitudes larger than the effect of any (local) governance variable can indicate the possibility that the reduction of deforestation in the 1st and 2nd period analyzed is mainly related to supra-municipal government interventions (most probably federal government policies at the time), such as the PPCDAm.

We agree that this is an important point. We have now added this explanation to the text in the following sentence “Similarly, broader patterns of deforestation were also likely to have been influenced by federal and state government interventions to reduce deforestation across the biome (e.g. PPCDAm) (West & Fearnside, 2021)” (lines 431-433). In addition, throughout the manuscript (see lines 61-64, 297-299, 363-366, 433-435, 484-486, and 415-416), we added the framework of proximate and underlying drivers of deforestation (Geist & Lambin, 2002) to demonstrate that the effects of local governance variables were not the most direct and important drivers of deforestation, but were still indirectly important to understanding deforestation processes. This explanation helps to explain why the effects of local governance were smaller than the effects of time period fixed effects and lagged deforestation. 

(Comment 3) : Another finding I think is very interesting is the positive association between deforestation rates and the implementation of municipal environmental agencies in the lagged model. Although it can be an effect of command and control policies (such as LPM) targeting municipalities with the highest deforestation rate, as already pointed in the manuscript. This result serves to warn local environmental governance studies to context-specific dynamics that may prove counter-intuitive. In my opinion, the authors could stress this a little more.

Thank you for the suggestion. We added an additional sentence to emphasize this point, as follows “In addition, by demonstrating that governance indicators may not always have the expected relationship with environmental outcomes, this result emphasizes the importance of considering context-specific governance dynamics that may influence theoretical relationships” (lines 528-530). 

(Question 1) : The authors surveyed different public databases but did not mention the IBGE Agricultural Census (Censo Agropecuário), in which there is information for the municipal level on land tenure, number of properties, and their sizes. Is there a particular reason why the authors have not included this in their model? In my opinion, one of the measures of local governance is related to the security of tenure and this is not present in the model.

Yes, we did thoroughly dig into the important data from the Agricultural Census and wanted to use this data source. However, given that the census was collected only in 2006 and 2017, there was not sufficient longitudinal data to include this data source for each of our time periods. We have now added a quick reference to this data source in the manuscript, as follows “We omitted all variables that were not available for at least three time periods (e.g. those from the IBGE Agricultural Census)” (line 275-276). 

Reviewer #2: The article presents results of a spatial panel fixed-effects regression model which relates publicly available governance and context variables from the Brazilian Amazon to deforestation in order examine relations between governance and deforestation at the municipal level. The article presents original new research, is overall well written and has a clear structure and line of argumentation that can logically be followed with conclusions that are based on the research findings.

The main draw back is a rather narrow perspective of the authors. They only have the Worldbank governance framework in mind and “their” municipality data. The study deserves and needs to be put into broader context. But this can be amended. A number of more specific issues are listed below. I recommend to publish this manuscript in PLOS One with some modifications and amendments.

Thank you for your review. We appreciate that you consider our manuscript to be well written, have clear structure, and draw appropriate conclusions. We are thankful for your thoughtful comments and suggestions, which significantly strengthened our paper. We address your concerns about the study’s framework and data as part of addressing your more specific concerns below. 

50 – 56 The article would benefit from a clear governance definition, you mention loosely actors and practices, but do not provide a definition. There are so many on the market. You have a focus on mostly governmental plus some economic issues only which is rather narrow in view of a modern governance understanding. Mention, explain and discuss this. Also as concerns good governance – there are many concepts. Intro in general: how is governance related to other deforestation drivers? Provide a conceptual understanding that will at the end help considerably to interpret your findings – some more details are given below.

Thank you for pointing out that we did not previously include a clear definition of governance. We have now added this to the manuscript in the following sentence “Forest governance–defined as "the set of regulatory processes, mechanisms and organizations through which political actors influence forest actions and outcomes (Agrawal et al., 2018)"– occurs across multiple spatio-temporal levels and scales, involving interactions between actors with different incentives, responsibilities, and practices related to use, management, and protection of forest areas and resources” (lines 55-60). 

We address your comments about our use of a relatively narrow view of governance below in our additions relating to our use of publicly-available data that was mostly not designed for governance assessments. The large scale of this analysis, the use of secondary data sources, and the adaptations to work within the constraints of a governance framework, together accounted for the more restricted conception of governance presented in this study. 

In addition, we provided conceptual information on the proximate and underlying drivers of deforestation. See your more specific comment below for the exact changes. 

67 There is research on subnational levels, see Nansikombi et al (Forest Policy and Economics 120 (2020) and Fischer et al (World Development 148 (2021)

Thank you for your suggestion. We have now added these citations, as follows “Relatively little research has focused on the impact of municipal-level governance on forest change, despite evidence that local-level governance is important and should be monitored by policymakers (Larsen, 2011; Secco et al., 2014; Nansikombi et al., 2020; Fischer et al., 2021)” (lines 78-80). We additionally now reference these citations in other sections of the manuscript. 

94-96 “Other studies have found that some governance indicators were correlated with negative outcomes for forests and the environment. For example, increases in indicators related to business and the economy“ – business and economy are not governance! e.g. Ceddia 2014 analyzes agriculture AND governance and their relations, but agriculture indicators are not governance.

We agree with your comment. We adapted the previous sentence to now express that Ceddia et al., (2014) analyzed both agriculture and governance in relation to deforestation. The new sentence now reads “Other studies have found that some indicators of governance were correlated with negative outcomes for forests and the environment. For example, in some situations where good governance reduced bureaucratic challenges facing private businesses, good governance was also associated with negative environmental outcomes, including higher deforestation (Ceddia et al., 2014; Evans et al., 2018)” (lines 109-113). As we describe in other sections of this document, we recognize that agriculture indicators are not governance, though can serve as proxies for regulatory quality as well as for the direct drivers of deforestation. This is a discussion we explore in further detail in the content we have now added to section 4.1 where we weigh whether our chosen indicators are better analyzed as direct drivers of deforestation. 

129 “where the official monitoring system of Brazil (PRODES) detected deforestation during 2005-2018 “ better write “for which deforestation data were available”, otherwise it reads as if you excluded municipalities with zero deforestation, which you probably did not do, right?

We have now made this change. The new sentence reads “Our study included all municipalities in the Amazon biome in Brazil for which data on deforestation were available from 2005-2018 through the official monitoring system of Brazil (PRODES) (n=457, Fig. 1)” (lines 147-149). 

168 Section 2.1.2 contains your results, all this is based on your evaluations and I recommend to include this as the first (still descriptive) subsection of you results.

We have incorporated your suggestion by moving what was previously section 2.1.2 to now be section 3.1 of the results. 

197 there are many more , e.g. ….

Kishor, N., Kenneth, R., 2012. Assessing and Monitoring Forest Governance:: A user’s guide to a diagnostic tool. In, Program on Forests. PROFOR, Washington, D.C., USA.

Davis, C., Williams, L., Lupberger, S., Daviet, F., 2013. Assessing Forest Governance: The Governance of Forests Initiative Indicator Framework. In. WRI, Washington, D.C., USA.

de Graaf, M., Buck, L., Shames, S., Zagt, R., 2017. Assessing Landscape Governance, A Participatory Approach. In. Tropenbos, EcoAgriculture, Wageningen, Washington.

We added these citations and the sentence now reads “The development of analytical governance frameworks has been instrumental for researchers and organizations to understand and systematically compare important characteristics of governance systems across diverse localities (e.g. Kishor & Belle, 2004; Kishor & Kenneth, 2012; Davis et al., 2013; Graaf et al., 2017; Wehkamp et al., 2018)” (lines 197-200). 

207 delete “perceptions of”

Done. 

237 this is really a main constraint. It also becomes clear that you more or less pick what is available and from this pragmatic end, but not from a scientific perspective you design your study. E.g. Kaufmann on whom you base your concept says “Of the 31 data sources used in 2009, 5 are from commercial business information providers; surveys and NGOs contribute 9 sources each; and the remaining 8 sources are from public sector providers.” It is well discussed later in your paper but make this clear on a prominent place (title or abstract and conclusions), talk about “selected”, or “government perspective” … or. You argue that you do not rely on perceptions but on measured data – ok, but the drawback is that you need to take what is there, interpret this to make it fit in your categories, instead of asking/assessing the hard and essential governance factors.

Thank you for this thoughtful comment. We have now addressed this concern in several sections of the manuscript. 

First, we have now expanded on our data collection process in section 2.2.1 of the methods. We aimed to clarify that our approach was not only to use data that was available, but to carefully sort through all available data to select those variables that best fit within the governance framework. The section now reads “In total, we identified over 105 potential variables from 17 sources that tracked changes in governance across a wide array of sectors, including public policy, law, commercial enterprises, and the environment, among others. We then trimmed this initial larger dataset to fit within the constraints of our analysis. We examined the definitions and data collection processes of each variable to identify which ones most closely aligned with each indicator definition. We then assigned relevant variables to each indicator category. We retained only a subset of the initial set of variables, selecting those that were most useful and relevant to the governance indicators. We removed those that were poorly representative of governance concepts, those that varied so significantly between years or across municipalities that we had reason to suspect errors, and those with a narrow temporal window (see section 2.3.1)” (lines 224-234). 

Second, we additionally added a table to the Appendix to further exemplify the range of data sources that we sorted through to end up with our set of variables (Table A1). 

Third, we added content to the abstract to make this clear and transparent upfront. See the sentence “Drawing on the World Bank Worldwide Governance Indicators (WGI) as a guiding conceptual framework, and incorporating the additional dimension of environmental governance, we identified a wide array of publicly-available data sources related to governance indicators that we used to select relevant governance variables” (lines 31-35). 

Fourth, we added the following sentence to the conclusion “We expect that future studies that leverage data sources specifically designed for governance assessments, rather than publicly-available data sources, may find even stronger relationships between governance and deforestation” (lines 682-685). 

258 there is no variable definition in the Supplementary, but would be interesting. E.g. “crop density” – you leave me alone with “(crops/km22) – PAM/IBGE “ what is this? At least two sentences in the supplementary to make sure the reader knows what is behind each of the indicators’ data, some info is there in the “Glossary”.

We have now added several additional terms to the Glossary that were previously missing, including the full definitions and sources of these terms. These include: agricultural sector, non-agricultural sector, crop density, cattle density, enterprise, and master plan. 

269-271 what is “original forest”, primary forest, secondary forest, any forest ? I do not find deforestation data on INPE 2020. Do you calculate the data yourself? How? based on deforestation maps, based on satellite data? This is the target variable so it deserves and understandable and complete description . “approximately conform to normality” does the model require normal distribution or not? Do your data fulfill the requirements, or not – how do you test this?

We have now adjusted this text to add the term “primary forest” and to clarify that this data was sourced from the PRODES Project platform from INPE (2019), which we now cite. This text now reads “We used official data on annual deforestation for all municipalities in the Brazilian Amazon, which was sourced from Brazil’s publicly available PRODES Project platform (INPE, 2019). We defined average yearly deforestation rate as the total square kilometers of primary forest cover cleared over each time period divided by the number of years considered, which enabled us to calculate one deforestation rate for each of the three time periods” (lines 286-290). 

We additionally have now adjusted the text about distributions to read “The deforestation data was strongly right-skewed and followed a log-normal distribution. Hence, we log-transformed the deforestation metric in all time periods to reduce the skew of the model residuals and improve symmetry” (lines 292-295). 

352 and following discussion

The discussion would benefit from a theoretical framework of how governance and other drivers are linked to deforestation. You already have Geist and Lambin in your reference list. Consider to introduce this as a framework in the Introduction. If you follow their idea of proximate/direct drivers and underlying causes, then you very obviously confirm this with your study; your context factors crops and cattle are the direct causes with “several magnitudes” stronger effects. Governance is underlying and thus much more complicated to show effects, also see Nanasikombi (2020) and Fischer (2021). If you apply this framework then it becomes clear that your indicators RQ ag. Companies, RQ non-ag. Companies, RQ ag. Employees, RQ non- ag. Employees must predominately be interpreted as direct driver indicators – not governance, as they mostly reflect the agricultural production in the area (even though they may as well reflect some regulative quality). In this sense I would be very cautious to claim that the two employee indicators (specifically with rather low p values) are a basis to claim that “local governance played a role in deforestation dynamics”. If you do not take them into account than you have environmental fund (negative), environmental agency (positive), and female mayor (negative) as remaining evidence (all with p<0.05 only) and I would interpret this more cautiously.

Thank you for these thoughtful suggestions. We agree that the Geist & Lambin (2002) framework is a useful addition to this paper. We have now added descriptions of this framework in the first paragraph of the introduction, methods, results, and discussion. We have made changes to interpret the regulatory quality variables more cautiously. See the following quotes, below:

Introduction: “Governance has been recognized as an underlying cause of deforestation by indirectly influencing the direct (proximate) drivers of deforestation (e.g. agricultural expansion) (Geist & Lambin, 2002; Nansikombi et al, 2020; Fischer et al., 2021)” (lines 61-64)

Methods (section 2.3.3): “We selected a set of time-variant control variables in line with previous research (e.g. Nepstad et al., 2009; Soares-Filho et al., 2014; Cisneros et al., 2015) to account for other direct and underlying drivers of deforestation (Geist & Lambin, 2002)” (lines 297-299). 

Results (section 3.2): “The indicators of environmental governance and regulatory quality each had two variables associated with deforestation, although the variables representing regulatory quality may have been heavily influenced by the direct drivers of deforestation (see discussion)” (lines 363-366). 

Discussion (first paragraph): “However, the variables representing regulatory quality may have also captured variation for the effects of more direct drivers of deforestation (e.g. agricultural expansion)” (lines 420-422) and “Our study therefore builds upon knowledge that both the direct drivers of deforestation and underlying drivers such as local governance contribute to deforestation (Geis & Lambin, 2002; Nansikombi et al., 2020; Fischer et al., 2021)” (lines 433-435). 

Discussion (section 4.1): “These variables may have been dually linked to agricultural expansion–a direct driver of deforestation–as well as the underlying driver of regulatory quality. As such, in this section we discuss the significance of expansion of agriculture in terms of both the direct driver and the governance indicator of regulatory quality” (lines 443-446) and “One such analysis would become possible with datasets that can more clearly distinguish between the effects of the direct agricultural drivers from the effects of the underlying governance drivers” (lines 484-486). 

Additionally, we have adjusted the claim that “local governance played a role in deforestation dynamics” to now read “several variables related to local governance played a role in deforestation dynamics” (lines 415-416). 

363 of course not silver bullet, direct drivers need to be tackled, but in all such measures governance may play a role – thus indirect driver, see above. 

We have now adjusted this sentence to read “our study also suggests that subnational governance alone will not be sufficient to tackle the complexity of forest loss” (lines 427-428). 

375 be much more cautious, see above

See the new sentence in response to your previous comment about the discussion, above. 

381 – 395 You see: now you are discussing the direct driver agriculture, not governance!

Thank you for pointing this out. We have responded to this concern by introducing the section with an explanation that in this dataset we cannot tease apart the effects of the direct driver and the underlying driver, and therefore we describe the effects of both. See the quote “These variables may have been dually linked to agricultural expansion–a direct driver of deforestation–as well as the underlying driver of regulatory quality. As such, in this section we discuss the significance of expansion of agriculture in terms of both the direct driver and the governance indicator of regulatory quality” (lines 443-446). 

396 – 407 and again: you are not discussing regulatory quality but the direct drivers – even though you try to link it to regulatory quality in the last sentence which is a bit artificial.

See our explanation and new added sections above. We aim to discuss the general trends of the direct drivers of deforestation and regulatory quality with an understanding that these trends may be influenced by both of these factors. This is also why we previously titled this section “expansion of the agricultural sector related to deforestation” rather than referencing the indicator of regulatory quality in the heading. 

471 – 473 yes

Thank you. 

474 – 489 – nicely written and I agree

Thank you. 

499 – 523 When discussing improvements in the Governance Framework you should show that you are aware of other frameworks, I gave some , see above. Then discuss why did you select this one? Others are designed completely different. There are many issues that are missing in the Worldbank framework compared to others.

We added the following sentences into the manuscript to address this concern, “Many frameworks have been developed and operationalized to advance understanding of the role of governance in environmental management, including Program on Forests (Kishor & Rosenbaum 2012), the World Resources Institute (Davis et al., 2013), and the International Union for the Conservation of Nature (Campese et al., 2016), among others. We chose the WGI framework to guide our study because it is widely used by practitioners and policymakers in the field of international development (Kaufmann et al., 2007). We are therefore able to enter a global conversation with implications for policy at scale” (lines 202-209). 

514/515 I do not understand what you want to say

We adjusted the previous sentence to now read “We furthermore observed that concepts such as social equity have not been included in many governance frameworks” (lines 605-606). 

538/539 -skip this because you did not show anything about national data availability.

We deleted this sentence. 

540-541 this could better be a subchapter on “methodological considerations” or alike, it has not so much to do with further research.

We changed the heading of this section to read “Methodological Considerations” (line 615). 

562 – 564 above you advocated that you use measured instead of perceived data, now you ask for perceived (interview) data. Perhaps both needed?

We added an additional half sentence that clarifies that both are important. The sentence now reads, “Future research that relies on interviews with local stakeholders in a cross-section of municipalities in the Brazilian Amazon may shed further light on the relationships highlighted in this paper, as the analysis of both publicly-available data and perceptions data will be important to understand the role of governance on deforestation” (lines 659-662). 

559 – 571 this is not only “future research” it has a lot of policy implications as well: you recommend to revise/amend public data collection/reporting, find other title.

We have now changed this section to be titled, “Directions for Future Research and Policy Implications” (line 629). 

574 – 576 this is of course true, I would nevertheless formulate more cautiously something like “found indications that m l governance matters” … and rather at the beginning mention the data base limitations by only using publicly available data that was mostly not designed for governance assessments, and: stronger statistical relations might be expected if the data could be improved.

We have adjusted this sentence to now read “Our research found indications that municipal-level governance matters to deforestation in the Brazilian Amazon, with implications for subnational governance in other countries with multilevel forest governance systems (lines 674-676). 

We additionally have added the limitations of only using publicly available data in the abstract, methods, and conclusion. See below. 

Abstract: “we identified a wide array of publicly available data sources related to governance indicators that we used to select relevant governance variables” (lines 33-35).

Methods: Existing sentence - “Given that our specific research goals did not include original data collection, but rather a synthesis of publicly available data, we adapted the framework as described below” (lines 213-215), and new sentence - “As such, we were unable to use a similar perceptions-based dataset, and we therefore relied on publicly-available reported data representing proxies of governance outcomes” (lines 218-220).

Conclusion: “We expect that future studies that leverage data sources specifically designed for governance assessments, rather than publicly available data sources, may find even stronger relationships between governance and deforestation” (lines 682-685). 

590 – 593 did you research on informal rules? Which indicator was this? If not, then you should not conclude on this. Rather this is another indicator that may be missing in the World bank framework and could be mentioned in the discussion on amending the framework

We moved this sentence up to the discussion (section 4.7). 

In general I am pretty sure that you are not the first one to study municipal level governance – here is the result of 10 min lit search, also use “multilevel governance” and “landscape level governance” search terms

Thank you for the suggested citations. We previously cited Secco et al. (2014) throughout the text, and we have now added Larsen (2011) (line 80), Velasco et al. (2020) (line 51), and Börner et al. (2014) (line 163). We did not add the citation for Rantala et al. (2014) since it does not focus on the municipal level and is a qualitative study of multilevel governance. We also did not change the wording of our text as we believe that our claims about the lack of studies on quantitative municipal-level governance and deforestation still hold true. See the following sentences “Relatively little research has focused on the impact of municipal-level governance on forest change, despite evidence that local-level governance is important and should be monitored by policymakers (Larsen, 2011; Secco et al., 2014; Nansikombi et al., 2020; Fischer et al., 2021). Most comparative quantitative studies that analyzed the impact of governance on forest cover focused on national-level governance (e.g. Kaufmann et al., 2009; Umemiya et al., 2010; Fischer et al., 2020). Studies at the municipal level have primarily been case studies examining governance processes that are difficult to standardize and compare across a large sample of municipalities (Piketty et al., 2015; Sattler et al., 2016). Only one study we are aware of conducted a cross-municipal analysis of deforestation outcomes and governance in Brazil, though no clear relationships were found (Dias et al., 2015)” (lines 78-87). 

Municipal environmental governance in the Peruvian Amazon: A case study in local matters of (in)significance; P. B. Larsen;

Management of Environmental Quality 2011 Vol. 22 Issue 3 Pages 374-385

Secco et al. Forest Policy and Economics 49 (2014) 57–71

Scale and context dependency of deforestation drivers: Insights from spatial econometrics in the tropics, R. Ferrer Velasco, M. Kothke, M. Lippe and S. Gunter

PLoS One 2020 Vol. 15 Issue 1 Pages e0226830

Multilevel governance for forests and climate change: Learning from Southern Mexico

S. Rantala, R. Hajjar and M. Skutsch

Forests 2014 Vol. 5 Issue 12 Pages 3147-3168

Mixing carrots and sticks to conserve forests in the Brazilian amazon: A spatial probabilistic modeling approach

J. Börner, E. Marinho and S. Wunder

PLoS ONE 2015 Vol. 10 Issue 2

---

## [Decision Letter · Decision Letter 1]

27 May 2022

What's governance got to do with it? Examining the relationship between governance and deforestation in the Brazilian Amazon

PONE-D-21-31445R1

Dear Dr. Benzeev,

We’re pleased to inform you that your manuscript has been judged scientifically suitable for publication and will be formally accepted for publication once it meets all outstanding technical requirements.

Kind regards,

Stephen P. Aldrich, PhD

Academic Editor

PLOS ONE

Additional Editor Comments (optional):

Thank you very much for addressing the reviewer's comments so thoroughly in your revision.

Reviewers' comments:

Reviewer's Responses to Questions

**Comments to the Author**

1. If the authors have adequately addressed your comments raised in a previous round of review and you feel that this manuscript is now acceptable for publication, you may indicate that here to bypass the “Comments to the Author” section, enter your conflict of interest statement in the “Confidential to Editor” section, and submit your "Accept" recommendation.

Reviewer #1: All comments have been addressed

Reviewer #2: All comments have been addressed

2. Is the manuscript technically sound, and do the data support the conclusions?

Reviewer #1: Yes

Reviewer #2: Yes

3. Has the statistical analysis been performed appropriately and rigorously? 

Reviewer #1: Yes

Reviewer #2: Yes

4. Have the authors made all data underlying the findings in their manuscript fully available?

Reviewer #1: Yes

Reviewer #2: Yes

5. Is the manuscript presented in an intelligible fashion and written in standard English?

Reviewer #1: Yes

Reviewer #2: Yes

6. Review Comments to the Author

Reviewer #1: The new version of the manuscript is greatly improved and fit for publication.

The authors answered all my suggestions and comments in a positive way. Also, I must point out that the improvements based on R2 suggestions also greatly benefited the quality of this version (congratulations for the nice suggestions by R2 and the authors work on it).

Reviewer #2: the comments have been very thoroughly been taken into account and incorporated in the new version of the text

7. PLOS authors have the option to publish the peer review history of their article (what does this mean?). If published, this will include your full peer review and any attached files.

Reviewer #1: **Yes: **Vitor Bukvar Fernandes

Reviewer #2: No

---

## [Editor Report · Acceptance letter]

14 Jun 2022

PONE-D-21-31445R1 

What’s governance got to do with it? Examining the relationship between governance and deforestation in the Brazilian Amazon 

Dear Dr. Benzeev:

I'm pleased to inform you that your manuscript has been deemed suitable for publication in PLOS ONE. Congratulations! Your manuscript is now with our production department. 

Kind regards, 

on behalf of

Dr. Stephen P. Aldrich 

Academic Editor

PLOS ONE